# Novel *SCN5A* p.V1429M Variant Segregation in a Family with Brugada Syndrome

**DOI:** 10.3390/ijms21165902

**Published:** 2020-08-17

**Authors:** Michelle M. Monasky, Emanuele Micaglio, Giuseppe Ciconte, Valeria Borrelli, Luigi Giannelli, Gabriele Vicedomini, Andrea Ghiroldi, Luigi Anastasia, Emanuela T. Locati, Sara Benedetti, Chiara Di Resta, Giorgio Casari, Carlo Pappone

**Affiliations:** 1Arrhythmology Department, IRCCS Policlinico San Donato, San Donato Milanese, 20097 Milan, Italy; michelle.monasky@grupposandonato.it (M.M.M.); emanuele.micaglio@grupposandonato.it (E.M.); g.ciconte@gmail.com (G.C.); valiborrelli91@gmail.com (V.B.); giannelli.luigi@gmail.com (L.G.); gabriele.vicedomini@grupposandonato.it (G.V.); EmanuelaTeresina.Locati@grupposandonato.it (E.T.L.); 2Stem Cells for Tissue Engineering Laboratory, IRCCS Policlinico San Donato, San Donato Milanese, 20097 Milan, Italy; andrea.ghiroldi@gmail.com (A.G.); anastasia.luigi@hsr.it (L.A.); 3Vita-Salute San Raffaele University, 20132 Milan, Italy; diresta.chiara@hsr.it (C.D.R.); casari.giorgio@unisr.it (G.C.); 4Clinical Genomics—SMEL, IRCCS San Raffaele Hospital, 20132 Milan, Italy; benedetti.sara@hsr.it

**Keywords:** Brugada syndrome, sudden cardiac death, genetic testing, mutation, variant, *SCN5A*, sodium channel, arrhythmia, channelopathy, human, family

## Abstract

Brugada syndrome (BrS) is diagnosed by the presence of an elevated ST-segment and can result in sudden cardiac death. The most commonly found mutated gene is *SCN5A*, which some argue is the only gene that has been definitively confirmed to cause BrS, while the potential causative effect of other genes is still under debate. While the issue of BrS genetics is currently a hot topic, current knowledge is not able to result in molecular confirmation of over half of BrS cases. Therefore, it is difficult to develop research models with wide potential. Instead, the clinical genetics first need to be better understood. In this study, we provide crucial human data on the novel heterozygous variant NM_198056.2:c.4285G>A (p.Val1429Met) in the *SCN5A* gene, and demonstrate its segregation with BrS, suggesting a pathogenic effect. These results provide the first disease association with this variant and are crucial clinical data to communicate to basic scientists, who could perform functional studies to better understand the molecular effects of this clinically-relevant variant in BrS.

## 1. Background

There has been an ever-increasing interest in Brugada Syndrome (BrS) ever since its first description almost three decades ago, due to its ability to cause ventricular tachycardia/fibrillation (VT/VF) and sudden cardiac death (SCD) in young and otherwise healthy individuals [1]. Accordingly, BrS genetics has gained widespread popularity and is currently a hot topic, as not even half of BrS cases can be molecularly confirmed [2], and all but the involvement of the *SCN5A* gene is fiercely debated [3,4,5]. However, even the roles of several specific variants within the *SCN5A* gene are disputed, with many listed as variants of unknown significance, and many are thought to result in pathologies other than BrS, such as atrial standstill, atrial fibrillation, left ventricular non-compaction, dilated cardiomyopathy, Long QT syndrome, idiopathic ventricular fibrillation, and heart block [6,7]. Some *SCN5A* variants found in patients undergoing routine genetic testing have never been described before in the literature, or are listed in popular genetic databases, such as Varsome [8], as benign or likely benign.

Novel mutations in the *SCN5A* gene and their likely causative role in BrS have been of recent interest [9,10,11,12,13,14,15], as well as new candidate genes [13,16,17]. Most studies have reported autosomal dominant inheritance with incomplete penetrance [18,19,20], with a few suggesting a recessive or X linked inheritance [21,22] and a possible involvement of mitochondrial mutations [23]. Variants in the *SCN5A* gene associated with BrS result in a loss of function of the voltage-gated sodium channel subunit (Na_v_1.5) [17,24,25].

It is necessary to better understand clinical genetics to increase the power of the diagnostic capability based upon genetics alone. Genetic testing is a far easier test to perform clinically than other diagnostic tests, such as an ajmaline challenge, which requires the patient to travel to a highly-specialized facility because of the high risks of the procedure [26]. Travel restrictions or the cost of an invasive procedure could impair the ability to diagnose this potentially fatal disease, which often results in sudden death in otherwise asymptomatic and seemingly otherwise healthy individuals. Genetic testing, on the other hand, can be performed in more remote areas, either with a blood or saliva sample, without serious risks related to the procedure.

In this study, the variant NM_198056.2:c.4285G>A (p.Val1429Met) in the *SCN5A* gene is characterized for the first time, generally and in a family with BrS, providing crucial human data that are the first step in advancing diagnostic capabilities.

## 2. Case Presentation

Written informed consent of human subjects included in this case series report was obtained for their participation in the study and for publication. The procedures employed were reviewed and approved by the local ethics committee. The study was conducted in accordance with the Declaration of Helsinki, and written informed consent of human subjects was obtained for their participation in the study and for publication. The procedures employed were reviewed and approved by the local Ethics Committee (approver number: M-EC-006/A, rev. 1 March 2013).

The proband is a 38-year-old female with a personal history of syncope. Both her father (at 46 years) and maternal grandfather (at 70 years) died suddenly (Figure 1). Her maternal uncle experienced an aborted cardiac arrest at the age of 36 years old, and was diagnosed with BrS due to the presence of the type 1 BrS ECG pattern identified elsewhere after the diagnostic workup for aborted cardiac arrest. Thus, the proband underwent an ajmaline challenge at our facility, which resulted positive (Figure 2). She then underwent an electrophysiology study (EPS), and was inducible for ventricular tachycardia/fibrillation (VT/VF) (Figure 2). An ICD was implanted.

Genetic testing of several genes described in BrS research literature (*ABCC9*, *AKAP9*, *CACNA1C*, *CACNA2D1*, *CACNB2*, *DSG2*, *GPD1L*, *HCN4*, *KCND2*, *KCND3*, *KCNE3*, *KCNE5*, *KCNH2*, *KCNJ8*, *PKP2*, *RANGRF*/*MOG1*, *SCN1B*, *SCN2B*, *SCN3B*, *SCN5A*, *SCN10A*, *SEMA3A*, *TRPM4*) by Next Generation Sequencing and confirmed by Sanger sequencing revealed the novel heterozygous variants NM_198056.2:c.4285G>A (p.Val1429Met) in the *SCN5A* gene (LOVD: https://databases.lovd.nl/shared/variants/0000673713#00018523) (Figure 3) and NM_201596.2:c.1880G>A (p.Arg627His) in the *CACNB2* gene (LOVD: https://databases.lovd.nl/shared/variants/0000673714#00024101).

### 2.1. Assessment of Family Members

The proband’s 59-year-old mother is asymptomatic. Due to her family history, she underwent an ajmaline challenge at our facility, which resulted positive (Figure 4). She was not found to be inducible for VT/VF at EPS. Genetic testing by Sanger sequencing was positive for the familial variant NM_198056.2:c.4285G>A (p.Val1429Met) in the *SCN5A* gene but negative for the familial variant NM_201596.2:c.1880G>A (p.Arg627His) in the *CACNB2* gene.

The proband’s maternal uncle experienced an aborted cardiac arrest and he came to our facility only for genetic testing, revealing by Sanger sequencing the presence of the familial variant NM_198056.2:c.4285G>A (p.Val1429Met) in the *SCN5A* gene in him as well. However, he was negative for the familial variant NM_201596.2:c.1880G>A (p.Arg627His) in the *CACNB2* gene.

### 2.2. In Silico Predictions

The novel c.4285G>A variant in the *SCN5A* gene was classified as likely pathogenic according to ACMG criteria and Varsome database accessed on 13 July 2020 [8,27]:
PM1 Moderate: Hot-spot of length 61 base-pairs has seven non-VUS coding variants (seven pathogenic and zero benign), pathogenicity = 100.0%, qualifies as hot-spot.PM2 Moderate: Variant not found in gnomAD exomes (good gnomAD exomes coverage = 51.3). Variant not found in gnomAD genomes (good gnomAD genomes coverage = 33.1).PP2 Supporting: 272 out of 342 non-VUS heterozygous missense variants in gene *SCN5A* are pathogenic = 79.5% which is more than threshold of 51.0%, and 426 out of 1915 clinically reported variants in gene *SCN5A* are pathogenic = 22.2% which is more than threshold of 12.0%.PP3 Supporting: Pathogenic computational verdict based on 11 pathogenic predictions from DANN, DEOGEN2, EIGEN, FATHMM-MKL, M-CAP, MVP, MutationAssessor, MutationTaster, PrimateAI, REVEL and SIFT vs. no benign predictions.


Table 1 demonstrates the mutational hot-spot in the region harboring the novel *SCN5A* variant described.

The novel c.1880G>A (p.Arg627His) variant in the *CACNB2* gene was classified as likely benign according to ACMG criteria and Varsome database accessed on 13 July 2020 [8,27]:
BS2 Strong: Observed in healthy adults: gnomAD exomes allele count = 8 is greater than the five threshold for dominant gene *CACNB2* (good gnomAD exomes coverage = 69.5).BP1 Supporting: seven out of eight non-VUS missense variants in gene *CACNB2* are benign = 87.5% which is more than threshold of 51.0%, and 79 out of 267 clinically reported variants in gene *CACNB2* are benign = 29.6% which is more than threshold of 24.0%.


## 3. Discussion

In the present study, we report for the first time the variant NM_198056.2:c.4285G>A (p.Val1429Met) in the *SCN5A* gene, both generally and in BrS. The family segregation analysis and the in silico predictions support the hypothesis of a pathogenic effect of this variant and provide the first step towards understanding the pathophysiology in these patients and improving diagnostic capabilities.

The clinical presentations of the family members presented are severe, ranging from cardiac arrest to spontaneous type 1 pattern, syncope, inducibility for VT/VF during EPS, and a family history of sudden death. Clearly, aborted cardiac arrest is the most severe presentation of the disease possible. However, the presence of both syncope and a spontaneous type 1 ECG pattern have also been associated with a poor prognosis (6%–19% of people experiencing an arrhythmic event within 24–39 months during the follow-up period) [16,28]. Inducibility for VT/VF during EPS is also indicative of a poor prognosis [29,30]. Therefore, the clinical phenotypes indicate the presence of a severe disease, potentially requiring ICD implantation to prevent life-threatening arrhythmias.

The *SCN5A* gene encodes for the alpha subunit of the Na_V_1.5 protein. Disease-causing variants found in the *SCN5A* gene responsible for BrS result in a loss of function of the Na_V_1.5 protein and reduced sodium transport due to any of a variety of mechanisms, including reduced expression [31,32,33], non-functional channels [34], and changes in gating properties [35,36].

The c.4285G>A variant in the *SCN5A* gene is currently listed as likely pathogenic [8] and has several factors supporting the possible pathogenicity of this variant, including its genic expression and apparent low frequency. In fact, to date, the heterozygous c.4285G>A variant has never been found in the GnomAD database, consistent with the hypothesis of having a very low frequency in the general population. Supporting the hypothesis of a pathogenic role of this variant, the interspecies conservation of the p.Val1429 residue is shown in Figure 3B, according to Uniprot. Several other variants in this gene are known to be either pathogenic or likely pathogenic. In particular, Table 1 shows at least seven of such heterozygous mutations localized very close to our variant. Therefore, the region from position 38598716 to 38598763 might be a hot-spot for pathogenic variants. Moreover, several computational studies and many bioinformatic tools predict this heterozygous variant to be pathogenic, while no computational studies or bioinformatic tools predict the variant to be benign.

The NM_201596.2:c.1880G>A (p.Arg627His) in the *CACNB2* gene was additionally found in the proband, but not in her mother, who exhibits the *SCN5A* familial variant and BrS. The proband’s father died suddenly at the age of 46 years old. Therefore, it is likely that the proband inherited the *CACNB2* variant from her father (as opposed to being a de novo variant). Some variants in the *CACNB2* gene are currently disputed as possibly causative for BrS. However, considering also the presence of the *SCN5A* variant in the proband in the current study, the family segregation information for the *CACNB2* variant is inconclusive.

## 4. Concluding Remarks

The novel heterozygous variant NM_198056.2:c.4285G>A (p.Val1429Met) in the *SCN5A* gene segregates with BrS in the family presented, suggesting a pathogenic effect of this variant. These crucial human data are the first step in understanding the pathology of BrS for patients with this variant and set the stage for both functional studies to better understand the molecular pathways involved, and eventually better diagnostic capabilities, based upon very minimally invasive and safe genetic tests.

## Figures and Tables

**Figure 1 ijms-21-05902-f001:**
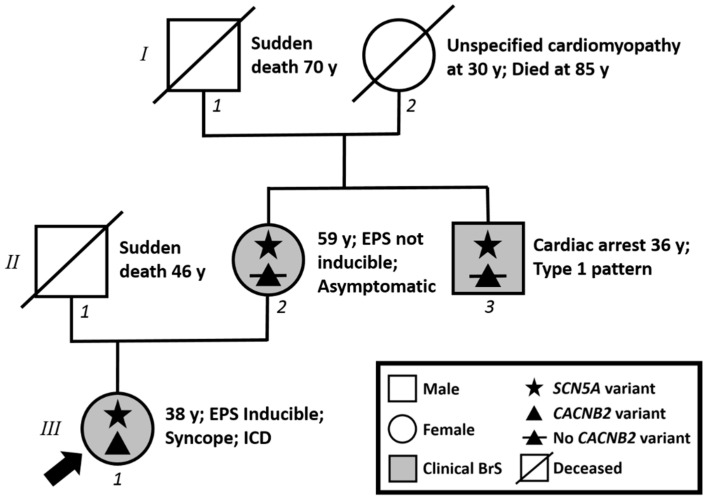
Family pedigree. Proband identified with arrow. Square: male; Circle: female; Shaded: clinically affected by Brugada syndrome; Star: molecularly confirmed *SCN5A* variant; Triangle: molecularly confirmed *CACNB2* variant; Triangle with slash: negative for *CACNB2* variant; y = years old at diagnosis.

**Figure 2 ijms-21-05902-f002:**
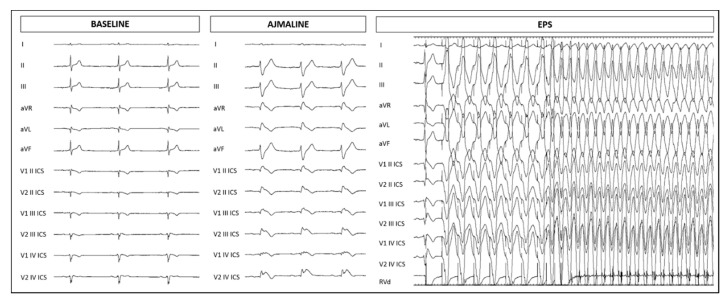
Electrocardiogram at baseline, after ajmaline administration, and ventricular tachycardia/ventricular fibrillation inducibility during electrophysiological study for proband.

**Figure 3 ijms-21-05902-f003:**
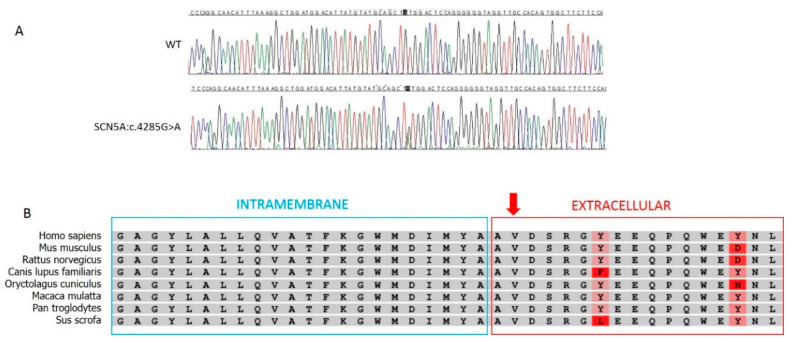
(**A**) Identification of the novel heterozygous variant c.4285G>A in the *SCN5A* gene by Sanger sequencing. (**B**) Interspecies conservation of the p.Val1429 residue, denoted by the red arrow (Uniprot annotation).

**Figure 4 ijms-21-05902-f004:**
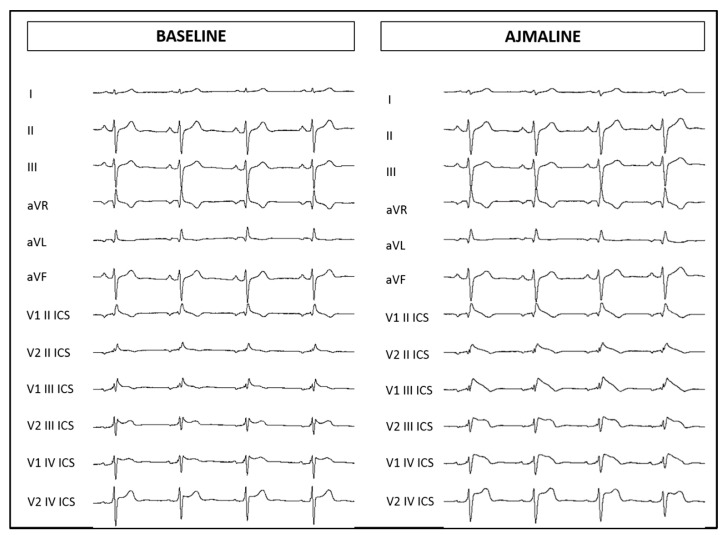
Electrocardiogram at baseline and after ajmaline administration for proband’s maternal aunt.

**Table 1 ijms-21-05902-t001:** Demonstration of mutational hot-spot in the region harboring the novel c.4285G>A *SCN5A* variant.

Chromosome	Position	Reference Sequence	Altered Sequence	Mutation	Effect
3	38598716	A	G	c.4137 + 6T >C	Uncertain
3	38598720_1insC	None	C	c.4137 + 1dupG	Pathogenic
3	38598725	C	A	c.4134G > T	Likely pathogenic
3	38598726	CT	GA	c.4132_4133delAGinsTC	Likely pathogenic
***3***	***38598736***	***C***	***T***	***c.4285G > A***	***Likely pathogenic***
3	38598738	G	A	c.4121C > T	Pathogenic
3	38598759	C	T	c.4100G > A	Pathogenic
3	38598762	C	A	c.4097G > T	Pathogenic
3	38598763	C	G	c.4096G > C	Pathogenic

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
