# Peer review of "Novel SCN5A p.V1429M Variant Segregation in a Family with Brugada Syndrome"

_ijms, 2020, doi:10.3390/ijms21165902_

Round 1

Reviewer 1 Report

The authors provide interesting human data on the novel heterozygous variant NM_198056.2:c.4285G>A (p.Val1429Met) in the SCN5A gene, and demonstrate its segregation with BrS, suggesting a pathogenic effect. The strength of this manuscript is a clear and well explained clinical data. The study is limited and it does not add anything to better understand the mechanisms underlying BrS. However, as the authors say, their results provide disease association with this variant and could be of interest to basic scientists who could perform functional studies to better understand the molecular effects of this clinically relevant variant in BrS.

The manuscript could be improved if the author provided information on possible other variants presents in the case individual or family members.

They show that the case’s father also died of sudden death. Could a variant inherited from the father affect the way the SCN5A mutation is phenotypically expressed in the proband?

A minor observation is that the manuscript could be improved with a schematic figure indicating where the mentioned hotspot and the mutation are in the protein structure in the channel.

Also, a methodological explanation on how the genetic analysis was done is lacking.

Reviewer 2 Report

The authors report a novel SCN5A variant in Familial Brugada syndrome. The manuscript was well written and interesting. 

1. In the 72nd line, I think "Figure 2" is appropriate, not Figure 3. If it is, Figure 4 must be the third figure and Figure 3 must be the fourth figure.

2. In Figure 4, a legend for figure 4B is missing. 

Author Response

1. In the 72nd line, I think "Figure 2" is appropriate, not Figure 3. If it is, Figure 4 must be the third figure and Figure 3 must be the fourth figure.

2. In Figure 4, a legend for figure 4B is missing.

Response:

Thank you for your time and your feedback. The figure labels have been corrected. Also, we are re-inserting the legend for figure 4 in the document that we are uploading… apparently there was a problem when the document got typeset. In the previous figure 4, now figure 3, the legend should read, “Figure 3. A. Identification of the novel heterozygous variant c.4285G>A in the SCN5A gene by Sanger sequencing.
B. Interspecies conservation of the p.Val1429 residue (Uniprot annotation).”